# Characterization of the Log-Normal Model for Received Signal Strength Measurements in Real Wireless Sensor Networks

**José M. Vallet García**

Finnish Geospatial Research Institute, National Land Survey, Geodeetinrinne 2, FI-02430 Masala, Finland; jose.vallet@nls.fi

**Abstract:** Using the classical received signal strength (RSS)-distance log-normal model in wireless sensor network (WSN) applications poses a series of characteristic challenges derived from (a) the model's structural limitations when it comes to explaining real observations, (b) the inherent hardware (HW) variability typically encountered in the low-cost nodes of WSNs, and (c) the inhomogeneity of the deployment environment. The main goal of this article is to better characterize how these factors impact the model parameters, an issue that has received little attention in the literature. For that matter, I qualitatively elaborate on their effects and interplay, and present the results of two quantitative empirical studies showing how much the parameters can vary depending on (a) the nodes used in the model identification and their position in the environment, and (b) the antenna directionality. I further show that the path loss exponent and the reference power can be highly correlated. In view of all this, I argue that real WSN deployments are better represented by random model parameters jointly accounting for HW and local environmental characteristics, rather than by deterministic independent ones. I further argue that taking this variability into account results in more realistic models and plausible results derived from their usage. The article contains example values of the mean and standard deviation of the model parameters, and of the correlation between the path loss exponent and the reference power. These can be used as a guideline in other studies. Given the sensitivity of localization algorithms to the proper model selection and identification demonstrated in the literature, the structural limitations of the log-normal model, the variability of its parameters and their interrelation are all relevant aspects that practitioners need to be aware of when devising optimal localization algorithms for real WSNs that rely on this popular model.

**Keywords:** received signal strength; wireless sensor networks; log-normal model; model identification; parameters' variability; parameters' correlation

---

## 1. Introduction

The classical path loss model typically used to characterize the average large scale loss and the shadowing of a signal traversing from a transmitter to a receiver can be written as ([1], p. 139)

$$PL(d) = \overline{PL}(d_0) + 10\,n \log\,(d/d_0) + \epsilon, \quad \epsilon \sim \mathcal{N}(0, \sigma^2) \tag{1}$$

where $PL(d)$ is the average path loss measured in dB as a function of the transmitter–receiver (T-R) distance $d$, log represents the logarithm in base 10, $\overline{PL}(d_0)$ is the path loss at a reference distance $d_0$, $n$ is a constant

denoted "path loss exponent" whose value depends on the specific propagation environment, and $\epsilon$ is a zero mean, normally distributed random variable with variance $\sigma^2$ accounting for the shadowing and other sources of uncertainty.

Equation (1) is said to be a *site specific* model in that its parameters $n$ and $\sigma$ have to be estimated from data collected in a specific site, and can then be used as a signal predictor in that site. In practice, and given the generality of the model, values of $n$ and $\sigma$ are reported for types of sites and reused in other similar environments. As an example, in a classic influential study, Seidel and Rappaport conducted an extensive measurement campaign in the early 1990s and provided deterministic values for $n$ and $\sigma$ for different types of buildings [2]. It is worth noting and emphasizing at this point that Equation (1) aims at modeling the path loss due to the signal traversing a more or less cluttered environment, and that in particular it does not include any HW related effects. In fact, the campaigns carried out to estimate $n$ and $\sigma$ in Equation (1) use high-quality professional radio equipment with highly omnidirectional antennas in order to minimize the effect of the HW in the experiments.

The signal strength measured by a receiver at a given T-R distance can be calculated in a straightforward manner by subtracting the path loss (Equation (1)) from the transmitted power and taking into account the transmitter and receiver gains, that is,

$$z(d) = \underbrace{P_{\mathrm{t}} + G_{\mathrm{t}} + G_{\mathrm{r}} - \overline{PL}(d_0)}_{z_0} - 10\,n\log\left(d/d_0\right) + \epsilon \tag{2}$$

$$= z_0 - 10\,n\log\left(d/d_0\right) + \epsilon, \tag{3}$$

where $z(d)$ represents the RSS, $P_{\mathrm{t}}$ the transmitted power, and $G_{\mathrm{t}}$ and $G_{\mathrm{r}}$ the gain in the internal circuitry, cables and antennas of the transmitter and receiver, respectively. The independent factors are typically collected under a constant term, $z_0$, which in fact represents the RSS measured at a distance $d_0$. Henceforth, I will denote $z_0$ as the reference power. In Equation (3), the effects of the HW are then explicitly incorporated into $z_0$. Note that, at least in theory, these effects do not affect the value of $n$.

Equation (3) is perhaps the most popular RSS-distance model used in the scientific literature, and it is pervasively used for localization in WSNs. It is so popular that sometimes it is simply called "the" log-normal (RSS-distance) model, and for simplicity we will call it so in the remainder of this article. It is a simple radial model inspired by the theoretical Friis formula and the empirical Okumura/Hata and COTS-231 models (which all share the same structure [1,3–8]), and whose parameters are chosen directly from experimental data. Its form is simple in that it hides all the complex details of the radio-propagation phenomena and predicts the behavior of the signal strength using an elementary equation. The theory, and experimental measurements to some extent, suggests using a line to model the mean and a normally distributed random variable to model "the rest". However, whether to use this algebraic structure is often decided based more on mathematical convenience rather than on the real capabilities of the model to correctly predict the RSS. All in all, the log-normal model is a mathematical empirical over-simplified and very convenient model, but not necessarily accurate.

Using the log-normal RSS-distance model in WSN applications adds another set of characteristic challenges. First, WSNs are typically formed by low-cost nodes/devices that can exhibit noteworthy differences in the radio transceivers, antennas and cables among different units. Furthermore, in inhomogeneous environments the particularities of the local surroundings can differently affect the communication channels over which the (potentially many) different pairs of nodes communicate. This HW variability and environmental inhomogeneity will necessarily impact the model parameters, whose value will now depend on the specific (usually a pair of) nodes used during the collection of training data for the model identification and on their particular positions during the experiment. Consequently, the log-normal model parameters naturally exhibit certain degree of variability in WSNs. Yet, once its parameters are

estimated, typically the same model is used to predict the RSS of signals sent by any node and received by any other pair in the network.

Despite the characteristic challenges related to using the classic log-normal model in WSNs, the parameter's variability and its significance has received little attention in the scientific literature so far. Clearly, the ultimate impact and relevance depends on the particular field/application domain in which the model is to be used. In the field of localization in WSNs, it has been experimentally shown in ref. [9] that (a) maximum likelihood (ML) positioning can be rather sensitive to the correct selection of the model parameters, and that (b) model identification approaches involving one or more than a pair of nodes can result in notably different positioning errors. These results grant relevance to the study of the model parameter's variability and its causes, at least in relation to the field of localization.

The main goals of this article are (a) to uncover and characterize this variability, and (b) to explain the main factors that influence it and how they do so. In order to do so, I provide discussion around and empirical evidence derived from the analysis of spatially rich data collected with the help of a robot acting as a mobile beacon in three different environments, as explained in Section 2. After presenting the details of the model identification procedure in Section 3, the discussion, analysis and contributions will be articulated around different sections, as explained hereafter.

It is well known that the log-normal model is misspecified, in that it does not include relevant explanatory factors of the underlying propagation phenomena such as the effect of particular obstacles. Part of this misspecification is due to the limitations imposed by its simple algebraic structure: being the model linear, it assumes the same RSS decay date at all T-R distances, and being radial it assumes that its predictions depend only on the T-R distance, and not on the signal's angle of departure/arrival neither on any angular dependency on the effect of the environment. Radiality and linearity (further discussed in Section 4) are then structural model limitations imposed by its algebraic form, which assumes a rather homogeneous HW and environment. This formal simplicity is part of the model's beauty and utility, but it ultimately comes at a price: The model is often incapable of properly explaining the observations, especially those collected in inhomogeneous environments.

The limitations of the log-normal model when it comes to properly represent real observations can induce systematic bias in the RSS predictions and different types of correlation in the model, namely (a) spatial correlation in the shadowing, a topic widely studied in the literature (see, e.g., [10] and the references therein), and (b), as shown in this article, correlation among the model parameters. The existence of this latter form of correlation invalidates the classical assumption that $n$ and $z_0$ depend only on environmental and HW related factors, respectively: They are interrelated due to the complexity of the underlying propagation phenomena and the structural limitations of the model.

Section 5 presents a quantitative empirical study showing how, in WSNs, the values of model parameters estimated using site specific training data can manifest a remarkable variability not only when collected in different sites, but also when retrieved in the same site with different nodes and/or in different positions. The study also reports the mean and standard deviation of the model parameters calculated from the experimental data for different tests and environments. In view of the presented experimental evidence, Sections 6 and 7, respectively, reexamine the effect of the HW and environment when the log-normal model is used in WSNs, emphasizing the effects that cannot be explained under the classical assumptions of (a) the parameters being deterministic values, and (b) $n$ and $z_0$ being independent. I further empirically show in Section 8 how $n$ and $z_0$ can be strongly correlated, and provide values for their correlation.

The evidence provided in this article challenges how the log-normal model is typically used in WSNs. First, the variability of its parameters suggests that these are better represented by random variables (RVs) rather than by deterministic ones. Furthermore, the correlation between $n$ and $z_0$ suggest that these shall be jointly distributed and jointly accounting for both, HW and environmental related factors. In view of all

this, in Section 9 I argue that the predictions can be more realistic if the model parameters are considered as RVs reflecting the above mentioned variability and dependence between $n$ and $z_0$. I then further argue that, as a natural consequence, incorporating these aspects into the measurements' generative process can lead to having more informative measurements, less biased and, accordingly, more plausible outcomes derived from studies relying on the resulting, better representative model.

While particular for the experimental setting used in this article, the statistics of the parameters reported here can be taken as examples of what practitioners can find in a real (but still laboratory controlled) WSN deployment, and as a guideline in further studies. The data and code necessary to reproduce the results shown in this article are openly available in ref. [11].

## 2. Experimental Setting

Let us begin by shortly explaining the experimental settings related to this article (for a more detailed description, see ref. [12], Section 1.4). In these settings we used a robot (see ref. [13]) as a mobile beacon. The robot automatically sent data packets with the maximum power allowed by the radio as it traversed different selected environments (see below). These packets were received by between 18 to 20 nodes of a WSN mounted in 1 m high poles and deployed in random fixed positions in the different environments, and their respective RSS were stored for off-line analysis. The robot is capable of simultaneous localization and mapping, and therefore its position was known at all times. Thus, the RSS measurements can be associated to the robot's position at any time instant. In addition, the poles of the fixed nodes appear as obstacles in the map generated by the robot, and therefore their position can be accurately estimated within a few centimeters' accuracy. This way, for each RSS measurement we have the transmitter and receiver position, from where the T-R distance can be calculated.

The robot and nodes of the network were equipped with TI CC2420 802.15.4 compliant radios using the 2.4 GHz ISM band. The communications protocol followed a token-passing scheme coordinated by a master node, in which each packet was assigned a unique transmission time slot. This eliminated the possibility of packet collisions and mutual interference. The protocol was so that the robot sequentially transmitted packets to each node using a predefined set of different free frequencies/channels at a rate of about ten packets per second. Thus, each receiving node produced approximately ten RSS measurements per second, each one corresponding to a different channel. The channel selection was manually done right before the experiments in each environment; after a scan of all available channels, only those with low noise were selected to be used during the experiments. For each respective node, the measurements were averaged over all channels and every 25 cm traversed by the robot. This reduced the effect of fast fading and standardized the density of effective measurements along the robot trajectory. The payload of the packets contained the set of RSS measurements and associated node identifiers corresponding to the precedent round of communications, and thus had a variable length of approximately between 30 to 50 bytes.

The experimental data was collected in three environments (see Figure 1) with different propagation characteristics representative of a broad range of scenarios: (a) A large basketball court (see Figure 1a) with a continuous clear line of sight (LoS), (b) the lobby of a building (see Figure 1b), a large open space more reflective and with some large and dominant obstacles (e.g., the stairs, large plant pots and a large metallic door), and (c) a typical modern office (see Figure 1c). Three independent tests each lasting around seven minutes were carried out in each of the environments. For each test, the robot followed a random trajectory aiming at roughly covering the same area of interest (AoI). (The reasons that justify what area and why is deemed as of interest are application specific and not relevant in this article. As an example, the reader can think of a reconnaissance mission in which the robot has to cover certain area collecting contextual information together with the rest of the nodes of the WSN.)

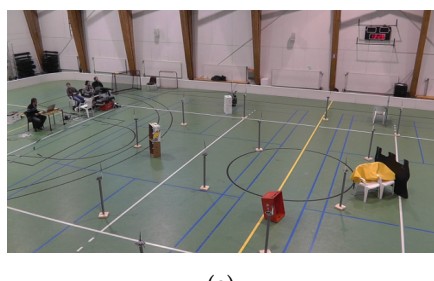 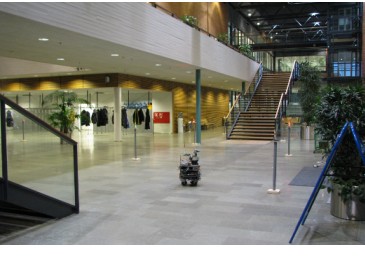 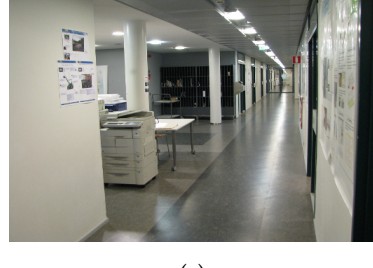

(**a**) (**b**) (**c**)

**Figure 1.** Environments of the experimental data collection campaign. (**a**) Basketball court; (**b**) lobby; (**c**) office.

The different maps corresponding to the three trajectories per environment were matched as described in ref. [14] to obtain a unique reference map and global coordinate system for each environment.

In summary, the data recorded consists of three sets of RSS measurements per environment collected by 18 to 20 nodes (the receivers) placed in well-known fixed positions, where each measurement can be associated to the position of the robot (the transmitter) thus allowing to calculate the true T-R distance.

## 3. Model Identification

Let us denote $\mathbf{z} = (z_1, \ldots, z_K)^t$ the vector of K random RSS observations made at T-R distances $d_k$, where $k = \{1, \ldots, K\}$ and the superscript $t$ denotes the transpose. Under the model (3) we can write

$$\underbrace{\begin{pmatrix} z_1 \\ z_2 \\ \vdots \\ z_K \end{pmatrix}}_{\mathbf{z}} = \underbrace{\begin{pmatrix} -10 \log (d_1/d_0) & 1 \\ -10 \log (d_2/d_0) & 1 \\ \vdots & \vdots \\ -10 \log (d_K/d_0) & 1 \end{pmatrix}}_{\mathbf{A}} \underbrace{\begin{pmatrix} n \\ z_0 \end{pmatrix}}_{\theta} + \underbrace{\begin{pmatrix} \epsilon_1 \\ \epsilon_2 \\ \vdots \\ \epsilon_K \end{pmatrix}}_{\epsilon} = \mathbf{A}\theta + \epsilon. \tag{4}$$

A full linear regression involves minimizing the sum of squared errors

$$J(\theta) = (\mathbf{z} - \mathbf{A}\theta)^t (\mathbf{z} - \mathbf{A}\theta). \tag{5}$$

The solution that minimizes (5) is the ordinary least squares (OLS) estimator

$$\hat{\theta} = (\mathbf{A}^t \mathbf{A})^{-1} \mathbf{A}^t \mathbf{z}. \tag{6}$$

We still need to calculate the parameter $\sigma$ in order to completely define the model. This can be estimated using the sample standard deviation of the residuals

$$\hat{\sigma} = \sqrt{\frac{1}{N-2} J(\hat{\theta})}. \tag{7}$$

Figure 2 shows examples of typical RSS-distance scatter plots and residual histograms together with their associated model for measurements collected in each of our three environments. Figure 2a shows a typical example for nodes in the basketball court. As we can see, the log-distance model is fit reasonably well to the trend of the RSS. The residuals are also close to being normally distributed, although in general they showed a tendency to be slightly skewed towards the negative RSS values. In the lobby the behavior was more heterogeneous. In general the log-distance model was still a good predictor of the radial average, but the residuals were often clearly not normally distributed. We observed the same type of skewness as in the basketball court, but often more accentuated. Figure 2b shows a representative example. Figure 2c

shows an example of a model from the office. We can observe a tendency of the slope of the average and the standard deviation to increase together with the distance. The residuals were rarely normally distributed, and often also multi-modal.

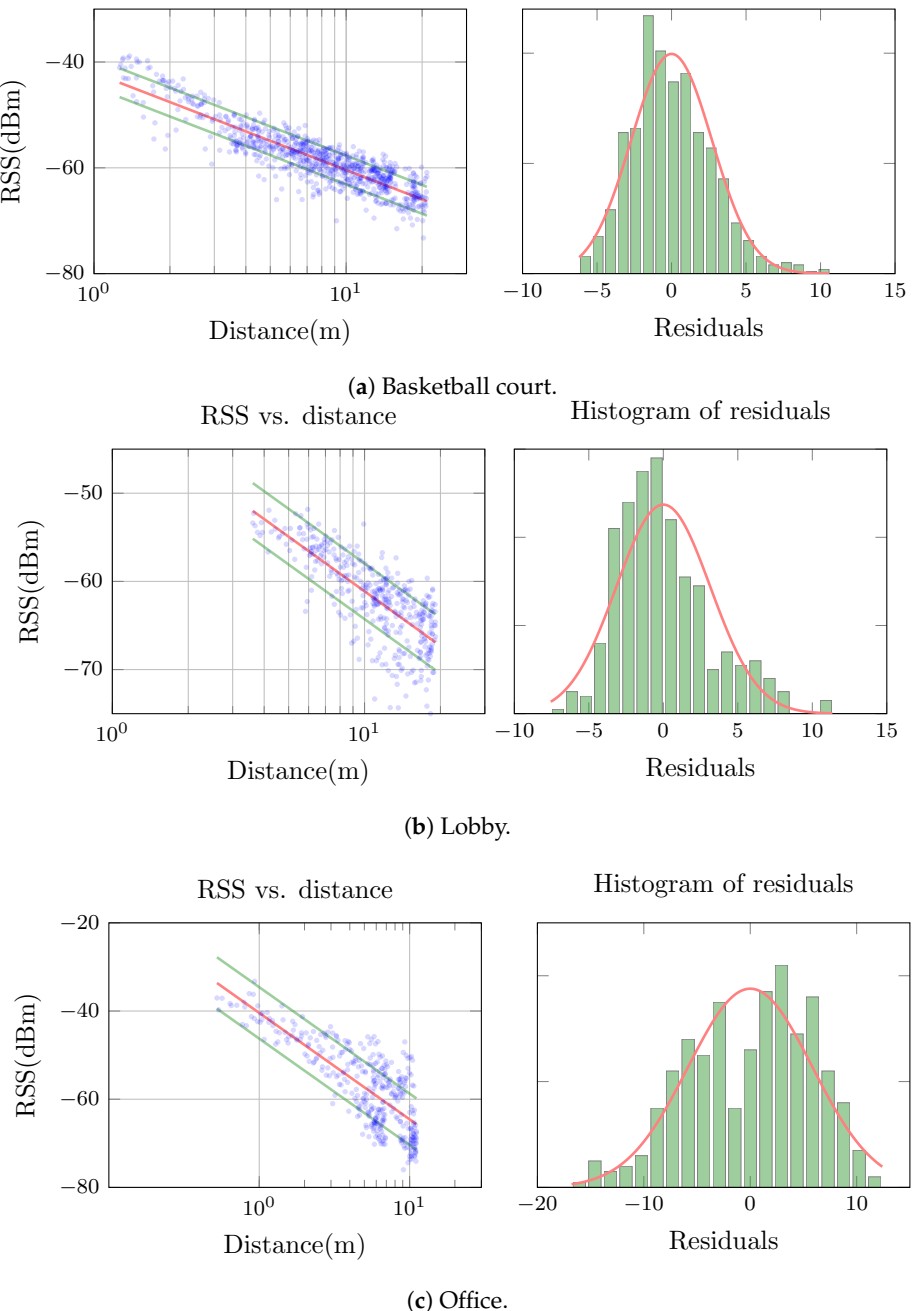

(**a**) Basketball court.

(**b**) Lobby.

(**c**) Office.

**Figure 2.** Examples of models identified in the different environments.

## 4. Structural Limitations

Let us now review some of the most relevant factors that limit the log-normal model's capability to accurately predict the RSS and how they affect the model itself. These are especially relevant when the log-normal model is used in WSNs.

## 4.1. Radiality

The log-normal model is radial, meaning that its predictions depend only on the T-R distance. This constraint makes radial models work optimally in homogeneous environments (e.g., with no obstructions) and when using omnidirectional antennas. In real WSN deployments the presence of obstacles is almost unavoidable. Moreover, low-cost and small size antennas are typically used, with radiation patterns that in practice exhibit show some degree of directionality. These facts represent challenges for a radial model, which cannot properly explain the observations. The model is then said to be misspecified, in that it lacks relevant explanatory factors of the underlying phenomena. As a result, $\epsilon$ in (3) becomes spatially correlated. This correlation needs to be taken into account by localization algorithms. Not doing so can further increase the bias and produce overconfident position estimates [15].

The impact of the log-normal model's radiality on its adequacy as a RSS predictor can be assessed in terms of the (relative) homogeneity of the environment and the omnidirectionality of the antennas.

### 4.1.1. Dependence on the Homogeneity of the Environment

Under the presence of obstacles, their geographical distribution, size and electrical characteristics impact the efficacy of the model to predict the RSS. One main factor that makes radial models suitable is then the homogeneity of the propagation environment. Very homogeneous environments will not favor any special direction of propagation on a macroscopic level. As a result, the net effect of the obstacles is a decay of the RSS as the T-R distance increases, with little or no dependence on the direction. On the other hand, inhomogeneous environments are more likely to produce favorable directions of propagation in the aggregate scale, making radial models less appropriate the more important this effect is.

WSNs typically operate in relatively small scales. For example, the maximum T-R distance achieved with our nodes is in the order of 20 m indoors. In these scales, a single dominant obstacle (e.g., a metallic wall) can significantly affect the propagation in the whole AoI. In general, one can argue that the propagation conditions can be heavily affected by the specific location of the nodes within the environment, simply because of the different relative distribution of obstacles. Therefore, in such a small scale networks the local characteristics of the environment can have a profound impact on the propagation channel and on the adequacy of the model.

Whether a radial model will or will not be useful depends on how well it can predict the observations within the AoI, and ultimately on the requirements of the application in which it is to be used. In a typical scenario in which the nodes' position of a WSN are to be estimated, the overall AoI is essentially the area covered by the WSN, which is the area for which the model has to explain the observations. The position of the nodes with respect to the AoI necessarily differs from node to node; still, the task of the model is to predict as well as possible the observations from all the nodes.

Because of the relative inhomogeneity of indoor environments given the scale of typical WSNs, the particular position of the nodes within the setup will uniquely condition the communication channel of each transmitter-receiver pair. Due to these heterogeneities, a single log-normal model will have to make global compromises in an effort to explain the observations from all the nodes. Instead of using one global common model (CM), in ref. [9] the possibility of using one model for each receiving node has been exploited, a paradigm denoted as using individual models (IMs). An IM needs to explain only the observations of its associated node. This model's spatial specialization results in better overall RSS predictions for all nodes when compared to using one single CM, and can then result in better position estimates [9]. It does not, however, completely solve the intrinsic misspecification problem of radial models when used in inhomogeneous environments.

### 4.1.2. Antenna's Directionality

Because of its radiality, the log-normal model assumes that the antennas used are perfectly omnidirectional. WSNs, however, are typically formed by low-cost small nodes whose antennas have some degree of directionality. The RSS then becomes a function of the relative orientation between the transmitter and receiver. The impossibility of the log-normal model to explain this results once more in biased RSS predictions which are globally compensated by the model. The net effect can be seen in the model parameters, which have to change to accommodate measurements for which the model is not explicitly prepared. Ultimately, this will contribute to the bias of position estimators relying on the model. In Section 6.1 we will see a quantitative study of how the directionality of the antennas can affect the model parameters.

### 4.2. Linearity

Another structural limitation of the log-normal model is its linearity with respect to the log-distance. Due to this linearity, the model assumes a constant decay rate of the RSS. However, already in early empirical studies characterizing the large scale path loss in cellular mobile radio, in cluttered environments the observations have sometimes been found to follow a pattern in which the decay rate of the RSS increases with the log-distance. There are several theoretical explanations for this. For example, in urban scenarios Xia explained it in terms of the obstruction of the Fresnel zones [16], and in cluttered indoors environments Devasirvatham explained it in terms of the absorption of the obstacles [17]. We have observed a similar pattern in our experimental scenarios, especially in measurements collected in the office. Figure 3 presents a scatter plot of RSS-distance measurement pairs associated with one of the nodes located in the office. We can observe that the cloud trend is slightly curved, and therefore a straight line (dashed green line) is not the best function to be fitted to the data. For a comparison, the figure also includes a fit of a third order polynomial on the log-distance (solid red line), which better explains the observed trend. The linear predictor has three differentiated areas in terms of its bias with respect to the trend: One with positive bias for small distances, a following one with negative bias and the last one with positive bias again. The linear model globally compensates for this, creating a model that is systematically biased in all distances.

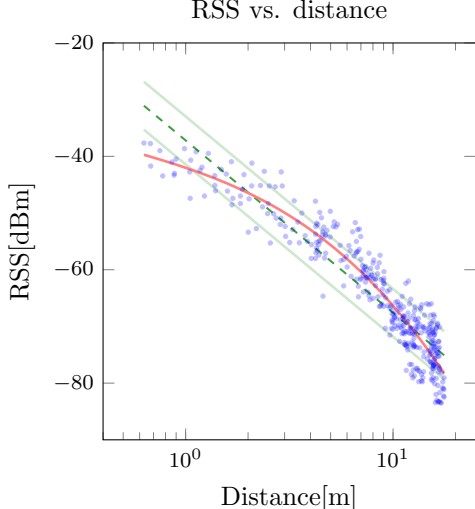

**Figure 3.** Received signal strength (RSS)-distance scatter plot for a set of measurements obtained in the office. The dashed greed line corresponds to a log-normal model fitted to the data, and the solid red line to a third order polynomial on the log-distance.

In addition to a systematic bias of the RSS predictions, the impossibility of the model to explain the curvature of the observation's trend induces correlation between the model parameters. This curvature can be caused by several factors. First, the log-normal model does not represent well what happens for rather short T-R distances. (It is worth remembering here that the Friis formula and the path loss model of Equation (1), and consequently the log-normal model of Equation (3), are valid only for the far field of the transmitting antenna, and for T-R distances much greater than the carrier wavelength). Being the transmitted power bounded, a RSS-distance scatter-plot should tend towards a fixed value as the T-R approaches zero. In the log-distance scale, this implies that the scatter-plot should become horizontal as the log-distance tends to $-\infty$. Therefore, the trend of scatter plots corresponding to real samples are curved at rather short distances. In practice, it can be the case that there are no situations in which the transmitter and receiver are in such short distances. However, if they do, this effect can become relevant in WSNs, as the the maximum communication distance is relatively short. Other evident factors that induce curvature in larger T-R distances are the clutterness and the inhomogeneity of the environment, which tend to increase the slope with the distance.

The interrelation between $n$ and $z_0$ in the log-normal model due to the effects of this curvature can be explained in terms of two antagonistic effects that will now be discussed in view of Figure 3. Note first that the reference point of the log-normal model is $(d_0, z_0)$ for any generic value of $d_0$, where $z_0$ depends on $d_0$. Typically, a value of $d_0 = 1$ m is used, and in that case the reference point corresponds to the ordinate in the origin (of the log-distance, that is, $d_0 = 10^0$).

Consider a node transmitting with the same power in different environments resulting in RSS-distance scatter plots with increasing curvature. The transmitted power being the same, and as argued before, in all cases the trend will roughly approach this transmitted power as the T-R distance decreases. Thus, the scatter plots can be imagined as being fixed at that point, where they all depart at short T-R distances. As the distance increases, the RSS trend will be lower the higher the curvature is.

When it comes to a log-normal model fit to these scatter-plots, there are two antagonist causes that affect differently the value of $z_0$. First note that the higher the curvature, the larger the value of $n$ will be. Now, on the one hand, an increasing curvature (increasing value of $n$) will reduce the average RSS in all T-R distances, and thus $z_0$ tends to decrease. On the other, being the model structurally limited by its linearity, an increasing curvature of the scatter plot will affect differently $z_0$ depending on the value of $d_0$. For $d_0$ close to $10^0 = 1$, $z_0$ will get a value close to that of the ordinate in the origin, which will clearly increase with the curvature (value of $n$). However, for relatively large values of $d_0$ (e.g., $d_0 = 10^1$ in Figure 3), the model linearity will induce a decrease in $z_0$ when $n$ increases.

The previously described effect necessarily induces correlation between $n$ and $z_0$. Moreover, this correlation should depend on the selection of $d_0$. As we will see in Section 8, this was indeed the case in our experimental setting, where we have observed correlation values of up to 0.9.

## 5. Variability of the Model Parameters

As argued before, in real WSN deployments the relative inhomogeneity of the environment due to the presence of obstacles can create unique propagation characteristics for each pair of communicating nodes depending on their specific location. Additionally, these networks are typically formed by many low-cost nodes, which, due to limitations in the manufacturing process, have slightly different radios even if they are the same model and from the same manufacturer. Moreover, the orientation of the nodes also affects the RSS, even when so-called omnidirectional antennas are used, as pure omnidirectional antennas do not exist. All these factors affect the model parameters.

The problem of dealing with differences in HW is sometimes referred to as the hardware variance problem in the existing literature. When the nodes are clearly different (different model and/or

manufacturer), it is intuitive and clear that the differences will affect the RSS. How to deal with this problem is under active research in the area of fingerprinting localization [18–21].

In the case of WSNs, being the nodes from the same manufacturer and commercial model, these small differences are usually ignored. Typically, the same RSS model is used for all the nodes, and no special measures are adopted to tackle the variability. Moreover, the models are identified using data collected with the same pair of nodes. The aim of this section is to provide an example of the variability of the model parameters that can be observed in real deployments depending on the particular pair of nodes used to identify the model and their position. This variability is intrinsically related to the HW variance problem in WSNs and to the relative inhomogeneity of the environment, relations that will be discussed in later sections.

Typically, one RSS-distance model is used by all the nodes in a network to predict RSS measurements. Recall that I denoted this type of model as a common model (CM). A site-specific CM is then a model whose aim is to predict the RSS optimally in a target environment or AoI for all the nodes involved.

Site-specific CMs are usually identified using RSS measurements obtained from a pair of nodes in the AoI, e.g., keeping one in a fixed position and placing the other at different locations. This is the typical procedure followed for example in cellular networks, in which the fixed node is the base station and the other one is the mobile terminal. In WSN, having many nodes to choose from and knowing that they are all slightly different, we can ask ourselves which pair should be used for the training data collection. Because the model is to explain observations from all the nodes, ultimately it would be desirable to estimate the model parameters using data from all the nodes.

In our experiments we have $N$ nodes in three different environments collecting RSS measurements in fixed positions of packets sent by a mobile robot. In each environment we have data collected in three tests in which the robot followed different trajectories (see Section 2). The best site-specific CM that optimally explains all the RSS measurements collected by all $N$ nodes during a single robot trajectory is the one whose parameters are estimated using all the $N$ data-sets (one set of RSS-distance pairs per node) associated with that test. In the following, I will refer to this model as the *reference model* for the given robot trajectory, and will use it as a reference to compare other models.

An alternative to using a CM to explain the measurements from all nodes is to dedicate one model to predict the observations from each of the nodes individually. These models are to be identified using only the measurements collected by its corresponding node. I denoted this type of model as an individual model (IM). Because the nodes are different and are placed in different positions, studying the parameter variability among IMs will give us an idea of the aggregate impact of the HW's variability and environmental inhomogeneity on the model parameters. I proceed now to do so using the data collected in our experimental setting.

Figure 4 presents the parameter values of the IMs and reference models identified using data collected in the three robot trajectories for each environment, together with empirical distributions for the parameters calculated with the data collected from each of the robot trajectories. The aim of the figure is to visually show the degree of variability in the model parameters that one can encounter in a real WSN deployment. The upper, middle and lower panels present information related to the path-loss exponent ($n$), the reference power ($z_0$) and the noise standard deviation ($\sigma$), respectively. In each of the panels, information associated to different environments is presented in different colors: red for the basketball court, green for the lobby and blue for the office.

The panels are divided in two parts separated by a thin black horizontal line placed at the origin of ordinates. The upper part of each panel presents empirical distributions of the corresponding model parameters, and thus the ordinates have units of probability density. The lower part presents the values of the model parameters used to calculate the empirical distributions of the upper part. These values are represented as dots, as explained below. The abscissa is common for both parts of the panel, and its tick values are presented in the base of the panel (below the lower part).

**Table 1.** Mean and standard deviation of the distributions from Figure 4, and parameters of the reference models (rm).

| | | **a Trajectory 1** | | | | | | **b Trajectory 2** | | | | | | **c Trajectory 3** | | |
|---|---|---|---|---|---|---|---|---|---|---|---|---|---|---|---|---|
| | | Bcourt | Lobby | Office | | | | Bcourt | Lobby | Office | | | | Bcourt | Lobby | Office |
| | mean | 1.71 | 1.70 | 2.87 | | | mean | 1.74 | 1.89 | 2.89 | | | mean | 1.72 | 1.39 | 2.84 |
| $n$ | std | 0.14 | 0.25 | 0.48 | | $n$ | std | 0.17 | 0.17 | 0.40 | | $n$ | std | 0.17 | 0.20 | 0.41 |
| | rm | 1.68 | 1.74 | 2.85 | | | rm | 1.73 | 1.91 | 2.92 | | | rm | 1.71 | 1.43 | 2.86 |
| | mean | −42.23 | −43.78 | −38.62 | | | mean | −41.93 | −41.67 | −38.58 | | | mean | −42.00 | −46.61 | −38.89 |
| $z_0$ | std | 2.20 | 2.87 | 4.65 | | $z_0$ | std | 2.5 | 2.78 | 3.75 | | $z_0$ | std | 2.7 | 2.27 | 3.80 |
| | rm | −42.51 | −43.43 | −39.11 | | | rm | −41.99 | −41.39 | −38.61 | | | rm | −42.09 | −46.06 | −38.98 |
| | mean | 2.65 | 2.98 | 4.76 | | | mean | 2.67 | 2.88 | 4.82 | | | mean | 2.66 | 2.46 | 4.86 |
| $\sigma$ | std | 0.28 | 0.47 | 0.71 | | $\sigma$ | std | 0.25 | 0.55 | 0.79 | | $\sigma$ | std | 0.23 | 0.26 | 0.83 |
| | rm | 3.00 | 3.40 | 5.34 | | | rm | 3.03 | 3.39 | 5.31 | | | rm | 3.05 | 2.85 | 5.38 |

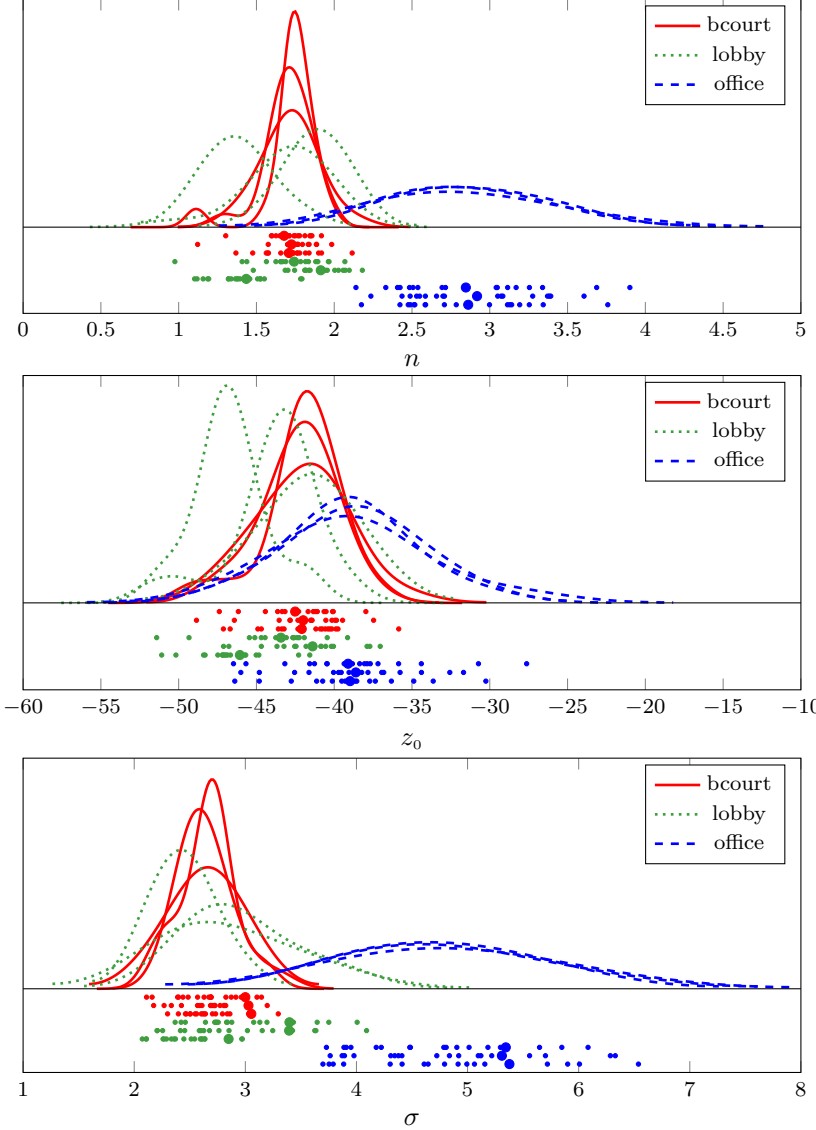

**Figure 4.** Reference (thick dots) and individual (thin dots) model parameters in the three environments for the three robot trajectories ($d_0 = 1$ m).

The lower part of the panel contains three sets of colored dots representing the parameter values of the reference model (thick dot) and the $N$ IMs (thin dots) corresponding to each of the three robot trajectories in each of the three environments. As explained before, red, green and blue dots correspond to models identified using data from the basketball court, lobby and office, respectively. The horizontal position (abscissa) of each dot represents the value of its associated parameter. The vertical position (ordinate) is used only to visually group the dots corresponding to the same robot trajectory: dots with the same vertical position belong to the same test/robot trajectory. Thus, altogether we can see nine sets of dots grouped by their vertical position: One per test/robot trajectory and environment. Correspondingly, there are nine associated empirical distributions in the upper part of the panels, one per group of dots/parameter values. These distributions were calculated using kernel density estimation as described in ref. [22] (code available in ref. [23]). The value of the reference distance was chosen as $d_0 = 1$ in all cases. Table 1 presents the main statistics of the distributions and the parameter values of the reference models.

Figure 4 shows in a visual and quantitative manner an example of the variability that the parameters of the classic log-normal model can experience in real WSN deployments not only depending on the type of environment, but also on the particular nodes used for the model identification and, through that, on their position in the (more or less inhomogeneous) environment. The general dependency on the type of environment is an issue that has received considerable attention in the scientific literature. However, the figure clearly shows that it is not the only relevant factor affecting the parameters, and that their values can vary significantly even within the same environment. This is clearly indicated by the dispersion of the groups of dots and, equivalently, by the variance of their associated empirical distributions. This variability is typically, if not invariably, ignored in the literature, and thus its consequences are not properly studied, assessed and addressed.

One of the main goals of this article is to uncover and characterize this variability, and to explain the main factors that influence it and how they do so. As explained before, these factors are the HW variability and environmental inhomogeneity, and Figure 4 presents their combined aggregate effect. Due to the complex interaction among the causes, accurately quantifying their individual contributions is quite complicated. In the following sections I then proceed to qualitatively argue about the effects of both in view of the evidence presented in Figure 4.

## 6. Effect of the HW

Consider a perfectly static, homogeneous and isotropic environment, like free space, and a group of perfectly equal nodes with perfect isotropic antennas. In such an ideal scenario, the values of all the model parameters would be equal for all the nodes and their densities would be Dirac delta functions. The path-loss exponent would depend solely on the environment, and in particular, in free space it would be exactly equal to two. The reference power $z_0$ would depend only on HW-related factors: The transmitted power, the gains of the transmission lines and antennas, etc. Finally, $\sigma$ would be small, and mainly due to thermal noise.

Now, consider using instead real nodes with small HW differences in the circuits, cables and connectors, and in general any internal circuitry that processes the RSS, but still with perfect omnidirectional antennas and in the ideal space. The net effect of these variability sources will be a different transmitted and received power among the nodes, which will be seen in every RSS packet independently of the T-R distance (studies in which the effects of transmitter and receiver variability on the signal strength are specifically measured are for example [24–26]). Because of this independence, the path-loss exponent is not affected, and only $z_0$ will be influenced. As a result, the dispersion of the probability distribution function (PDF) of $z_0$ will increase. Following this line of reasoning, in practice the

HW factors are considered to affect only $z_0$ and not the path-loss exponent. On the other hand, the path-loss exponent is usually assumed to be dependent on environmental factors only.

Regarding $z_0$ in Figure 4, we observe the following. First, the average depends on the environment. This is clearly related to the fact that we use $d_0 = 1$ m as the reference distance, distance at which the RSS is different in each environment due to the different values of $n$. Second, from a visual standpoint, the dispersion of the empirical distributions of $z_0$ change relatively less among environments than for $n$ and $\sigma$. This is confirmed by the statistics presented in Table 1, where we see that the relative variation of the standard deviations when changing environments is larger for $n$ and $\sigma$ compared to $z_0$. This suggests that $z_0$ is less affected directly by the environment than the other model parameters, which is in agreement with general practice.

Still, there is some variability that cannot be explained on the basis of HW effects only. First, although not being the main affecting factor, the change of environment does change the dispersion of $z_0$ with a clear pattern: It increases as the environment becomes more cluttered (see std. values in Table 1). Second, parameter values associated with the same node are different even among robot trajectories within the same environment. To this effect, compare in Figure 4 the dots presented along different lines belonging to the same environment. These relatively minor variations can be partially explained by the non-ideality of the antennas, the environmental inhomogeneity and the correlation between $n$ and $z_0$ induced by the linear structural limitation of the log-normal model, issues that are discussed in the following sections.

### 6.1. Impact of the Antenna Directionality

Let us return for a moment to our semi-ideal setup with different nodes and a perfectly homogeneous and isotropic environment, and let us add non-perfectly omnidirectional antennas, making it HW-wise more realistic. The received power will now depend not only on the relative T-R distance, but also on the orientation of both, the transmitter and the receiver. Then, a mobile receiver will in general not measure the same RSS in points located at the same distances. Therefore, in principle any model parameter could experience a change, although obviously we expect some parameters to be more affected than others. One immediate consequence is that $\sigma$ has to be increased to fit the observed data. This increase is not necessarily the same in all of the IMs, and therefore the dispersion of the empirical distribution of $\sigma$ will also increase. The effect of the antenna directivity on $z_0$ and $n$ is not so straightforward to evaluate in a general scenario; however, following the previous line of reasoning, we expect that their distribution will also be more disperse. This could in part explain the variability of $z_0$ which could not be attributed exclusively to the other HW-related factors.

The previous paragraph implies that, although not to the same extent, the directivity of the antennas can influence all of the model parameters. We will now seek to validate this statement using our experimental data. Quantifying this effect is very complicated in practice, as it would require being able to repeat the same experiments in the same locations with different directional antennas with known radiation pattern. Instead, we will incorporate offline the effect of different simulated antennas as if they were mounted on the robot during our experiments, and then we will compare the results.

The procedure is as follows. Imagine that we have an antenna with known pattern whose effect we want to evaluate. We know the position and orientation of the robot at all times together with the position of all the nodes, and thus we can calculate the relative node-robot angle for each of the RSS measurement from all the nodes. Using this information, we can calculate the antenna gain that would correspond to each measurement had the antenna been present during the experiment. We then add this gain to the respective observations. We do this for all the different antennas that we want to evaluate.

The previous procedure is not strictly equivalent to using a real antenna in practice. The main reason is that it implicitly assumes that the signals arrive to the receiver following only one path, the straight

line connecting transmitter and receiver, while it ignores the different antenna gains for the different multi-path components. The aim, however, is simply to obtain a first impression about how much the antenna directivity can affect the model parameters, and not to accurately quantify this effect. I then consider that the proposed method is suitable for the pursued purpose.

Instead of considering completely different antennas, we create one virtual antenna with a directivity that can be parametrically controlled. Our virtual antenna has the following radiation pattern:

$$g = g_{\mathrm{m}} + (g_{\mathrm{M}} - g_{\mathrm{m}})e^{-\kappa\phi^2}, \ \phi \in \{-\pi, \pi\}, \tag{8}$$

where $g$ is the gain in dB, $g_{\mathrm{M}}$ and $g_{\mathrm{m}}$ are the absolute maximum and minimum gains of the antenna, $\phi$ is the relative angle and $\kappa$ is the parameter that controls the directivity. Figure 5 show the radiation patterns for fixed values of $g_{\mathrm{M}} = 0$ dB and $g_{\mathrm{m}} = -30$ dB, and $\kappa$ varying from $10^{-2}$ to $10^1$. As we can see, the pattern becomes more homogeneous as $\kappa$ decreases. In the limit when $\kappa = 0$, the antenna is perfectly omnidirectional with a gain of 0 dB, and thus it will have no effect whatsoever. As the value of $\kappa$ increases, the antenna focuses more in directions close to zero degrees. Note that, because the gain will always remain between $g_{\mathrm{M}}$ and $g_{\mathrm{m}}$, the radiation pattern does not have any null, and therefore our antenna never blocks any direction completely; its effect is to simply decrease the power in certain directions. In the other limit of $\kappa = \infty$, the antenna has 0 dB gain for $\phi = 0$ exactly and $-30$ dB for any other direction. Therefore, it is almost omnidirectional, with the exception of a singleton direction whose probability of occurrence is zero. In this limit case, the antenna will, in practice, decrease all the RSS measurements by 30 dB, and that should be the only perceptible effect.

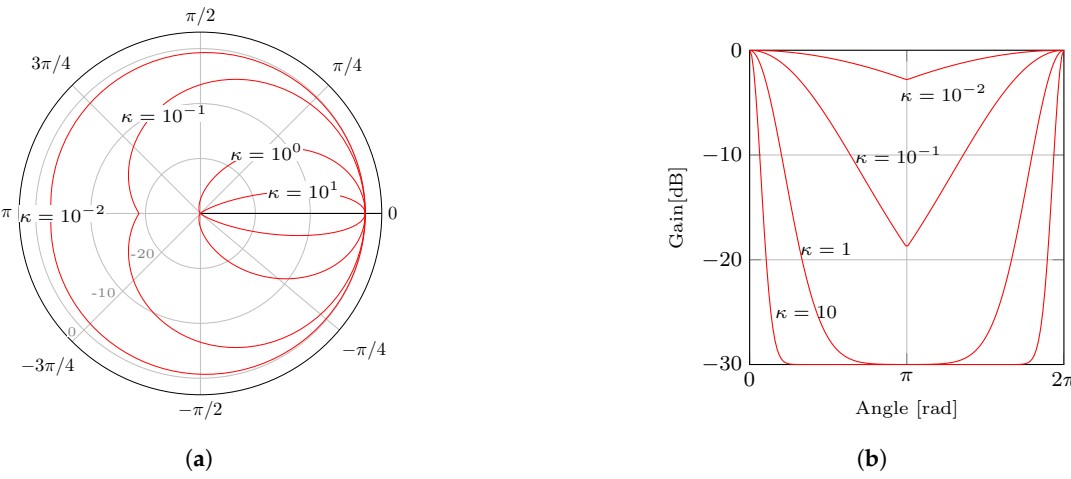

(**a**)                                                                        (**b**)

**Figure 5.** Antenna pattern for different values of $\kappa$. (**a**) Polar plot; (**b**) gain vs. angle plot.

Figure 6b shows the result of adding the effects from our virtual antenna with $\kappa = 1$ to the data collected during one of the robot trajectories in the basketball court. The figure compares the empirical distribution of the IM parameters when using the original and modified datasets. We see straightaway three clear effects attributable to the addition of the artificial antenna: The notorious increase and decrease of the averages of $\sigma$ and $z_0$, respectively, and an increase of the dispersion of $n$. Figure 6a shows an example of the effect of an antenna with $\kappa = 0.1$ on the RSS-distance scatter plot for a node in the basketball court. The relative variations of $n$, $z_0$ and $\sigma$ in this particular example were of 8%, 20% and 180%, respectively.

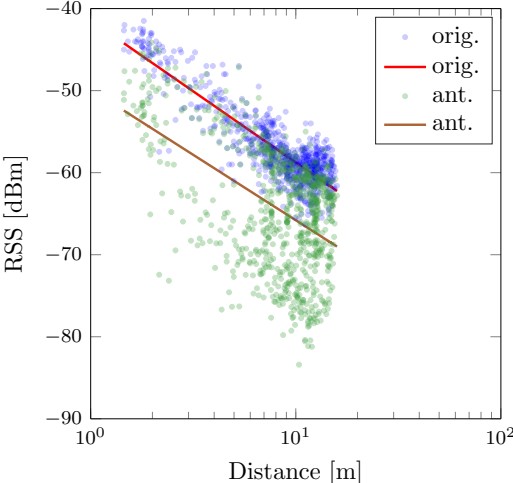

(**a**) Comparison of RSS-distance scatter plots for a node in the basketball court without and with a directional antenna ($\kappa = 0.1$).

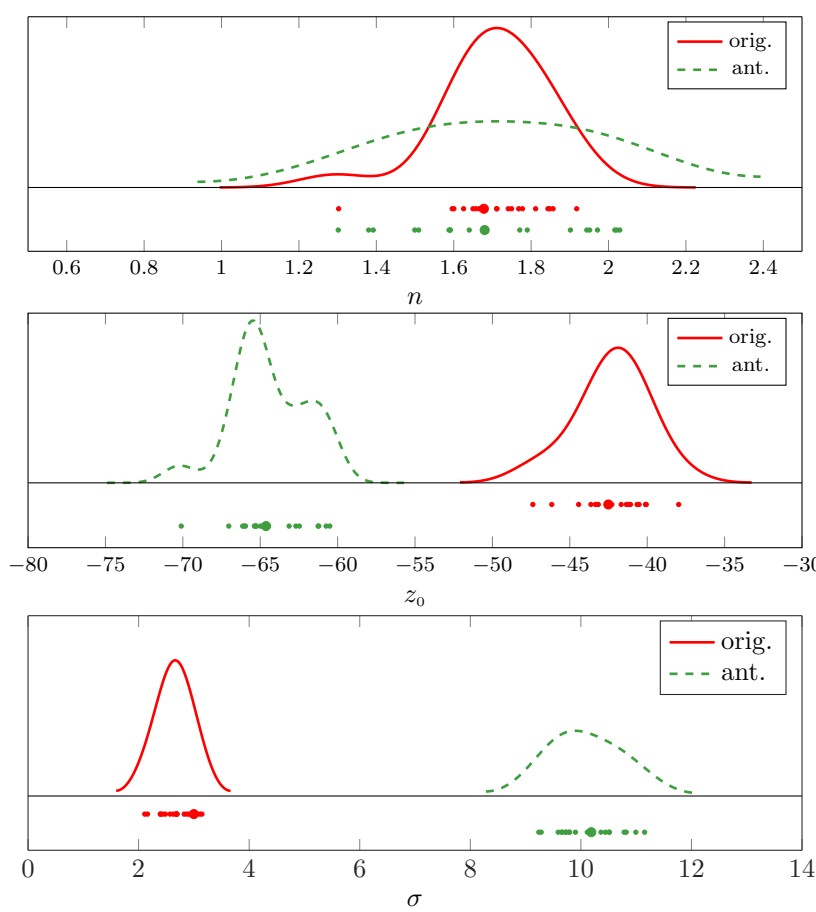

(**b**) Example of the effect of a directional antenna ($\kappa = 1$) on the parameters of individual models (IMs) (Orig. = original distribution, Ant. = distribution including the antenna effects). Data collected in one robot trajectory. $d_0 = 1$ m.

**Figure 6.** Examples of the effect of the antenna directivity in individual models (IMs). Data collected in the basketball court.

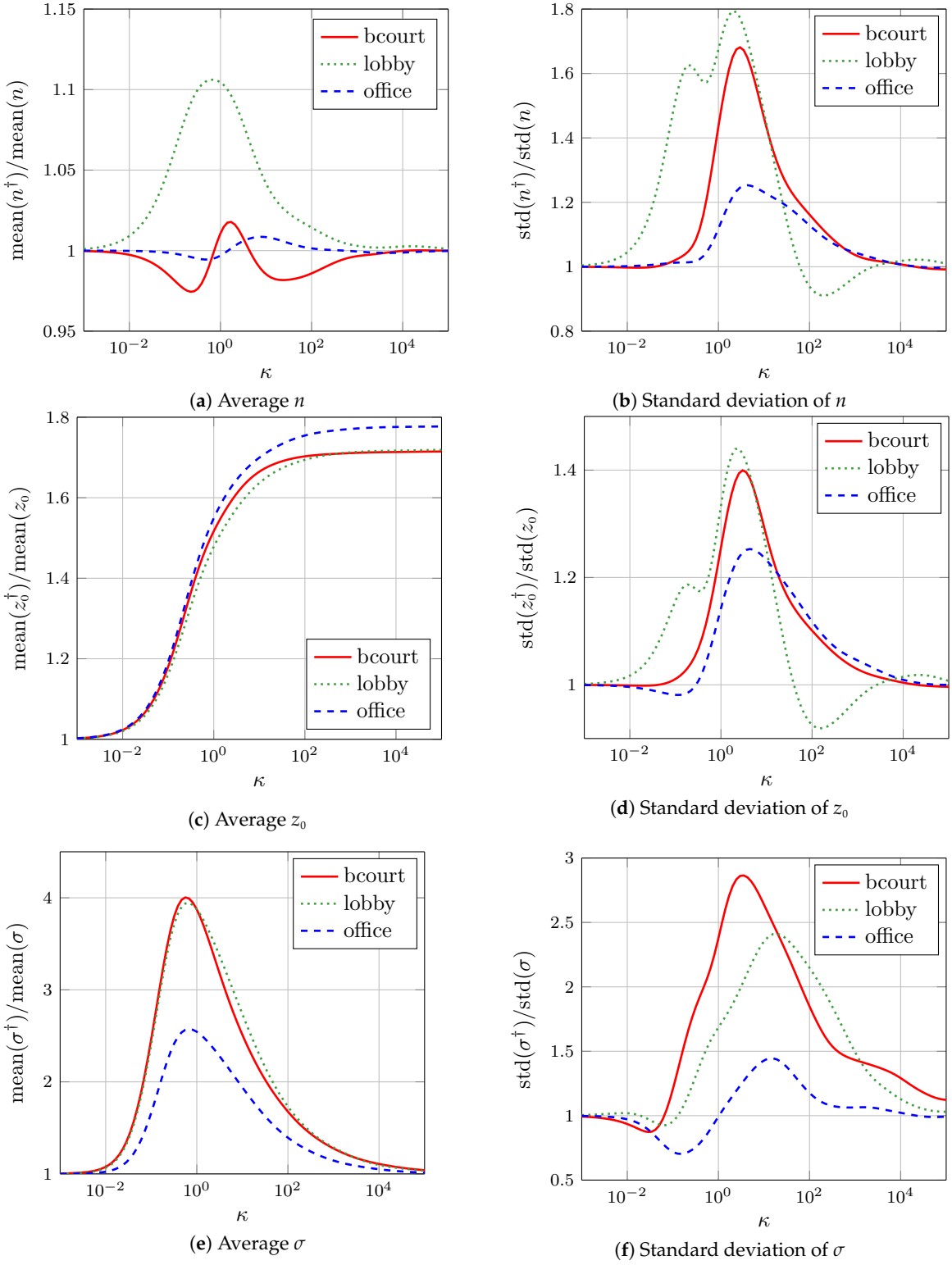

**Figure 7.** Relative increments of the mean and standard deviation of the model parameters' empirical distributions as a function of antenna directivity. Parameters with † refer to models with the addition of a virtual antenna, and those without it to the original models.

Figure 7 shows the effect of a progressive change of the antenna directivity (value of $\kappa$) on the empirical distributions of the IMs' parameters associated to one test/robot trajectory in each of the three experimental environments. The vertical axes show the relative changes of the mean and standard deviation of the model parameters' distributions with respect to their respective values without the effect of the virtual antennas. At first glance, we see that, in general, the changes in the parameters' distribution depend on the type of environment. Figure 7a shows that the change on the average path-loss exponent was about 2%, 11% and 1% in the basketball court, lobby and office, respectively. This suggests that the average path-loss exponent is not strongly affected by the antenna directivity. Its dispersion, however, can be strongly altered, as evidenced by the variations of its standard deviation shown in Figure 7b. These were of about 68%, 80% and 25% in in the basketball court, office and lobby, respectively. This suggests that the antenna directivity has a stronger relative impact on the path-loss exponent in open homogeneous environments. When the environment is homogeneous, the major source of uncertainty is the HW variability. This explains the larger relative importance of the antenna directivity in the basketball court and lobby.

The variation of the average $z_0$ was as expected given the behavior of our virtual antenna: a null effect for smaller $\kappa$ values and a decrease of 30 dB for larger ones (see Figure 7c). Note that the figure presents relative changes, and that $z_0$ is negative. Therefore, a decrease in the relative average implies an increase in the magnitude of the relative average. The dispersion of $z_0$ can also change with the antenna directivity (see Figure 7d). The maximum variations registered were of about 40%, 44% and 25% in the basketball court, office and lobby, respectively.

The average and standard deviation of $\sigma$ increased together with the directivity, as was expected (see Figure 7e,f). The strongest relative average increase appeared in the basketball court and lobby, where the relative contribution of the HW variability to the model uncertainty is the highest among our environments.

## 7. Effects of the Environment

The average and dispersion of the distribution of the path-loss exponent are largely affected by the clutteredness and homogeneity of the environment. As expected, in our experiments on average $n$ was higher in obstructed environments (office) than in more open ones (basketball court and lobby) (see Figure 4 and Table 1). The dispersion of $n$ can be explained in terms of the different robot-node LoS conditions associated to each node during the experiments. Broadly speaking, the higher the number and the greater the severity of the obstructions present between a particular node and the robot, the greater the slope (and perhaps curvature) of the scatter-plot trend will be. Accordingly, $n$ will be larger as well. In this way, varying LoS conditions experienced by different nodes will result in variability in the path-loss exponent among their corresponding IMs.

Different trajectories might then result in different average values and dispersion of the empirical distribution of $n$, as shown in Figure 4. We can see that, for different trajectories of the robot within the same environment, the densities for $n$ in the lobby are more separated than in the basketball court and office. As previously argued, the inhomogeneity due to the presence of dominant obstacles partially accounts for this difference. The lobby being a semi-open space, the average values of the path-loss exponent are more similar to those attained in the basketball court with respect to those obtained in the office. Moreover, because it is more inhomogeneous than the basketball court, in general the dispersion is larger. The office is also inhomogeneous, but more cluttered; thus it exhibited larger values of $n$ on average and with a great dispersion. However, the robot moved through the office along rather similar trajectories, as it had to follow corridors and move around in the few free spaces available. Thus, the densities were similar despite the inhomogeneity of the environment.

The impact of the antenna directivity on the path-loss exponent is, in general, clearly less significant than the effect of the environment. As an example, ref. [27] reported path-loss exponent changes of 10%

percent when switching from omnidirectional to directional antennas in the same room. This is also confirmed in our experiments: The largest relative change in $n$ among IMs shown in Figure 6b is in the order of 20%, whereas the relative changes in the reference models of Table 1 when switching from the basketball court to the office were of about 68%.

As discussed previously, the distribution of $z_0$ is most strongly affected by the HW particularities of the nodes. From the middle plot of Figure 4, we can see that the strongest effect resulting from a change of environment was a change of its average. Note that the averages are smaller in the lobby than in the basketball court, but are greater in the office than in the basketball court. As discussed in Section 4, this is caused by the structural limitation of the model to be linear and the selection of $d_0$, which can result in $z_0$ increasing with $n$. Had we selected a value of, e.g., $d_0 = 5$, the averages would be largest in the basketball court, followed by the lobby and finally by the office (see ref. [12], p. 50).

Varying LoS conditions can also affect the dispersion of the reference power $z_0$. To illustrate this, consider the situation in the lobby, a semi-open space with some large dominant obstacles like stairs (see Figure 1b). Depending on the relative position of the nodes and the trajectory of the robot, for some nodes there might be a clear LoS with the robot for longer time than for others. The two extreme cases would be that, given the same trajectory of the robot, for one node the stairs are always in between the node and the robot, whereas for the other there is always a clear LoS. Because of the continuous blockage by the stairs, in the first case the RSS will be more negative at all distances than in the second one, and therefore $n$ will be larger. In other non-extreme cases, the situation will fall in between, resulting in varying values of $n$. Now, as already argued, the variability in $n$ induces changes in $z_0$ throught their correlation. Thus, varying LoS conditions will also induce changes in $z_0$. This can be another factor that explains the increase in the dispersion of $z_0$ in the office compared to the basketball court and lobby (see std. values in Table 1)

From the lowest plot in Figure 4, we can see that $\sigma$ takes smaller and less disperse values in the basketball court. This means that the models have a relatively low uncertainty with respect to the models used in the other environments. In the office, $\sigma$ shows significantly higher and more disperse values. A relatively short dispersion suggests that all the IMs have similar level of uncertainty, whereas a lengthier dispersion indicates that the uncertainty varies a great deal among models. As discussed at the beginning of this section, this can be explained as being the result of having nodes with very different LoS conditions in relation to the robot trajectories. Clearly, $\sigma$ will be larger in deployments with variable LoS conditions.

## 8. Correlation Between n and $z_0$

As explained in Section 4, the model parameters can be strongly correlated. Figure 8a shows a scatter plot of the values for $n$ and $z_0$ from the IMs identified from our experimental data for $d_0 = 1$, where the correlation is evident. The value of the correlation coefficient depends on the selected value of the reference distance $d_0$. Table 2 presents this value for $d_0 = 1$ and $d_0 = 5$. Figure 8b displays its value as a function of $d_0$ for our three experimental environments.

**Table 2.** Correlation between $z_0$ and $n$.

| $d_0$ | Bcourt | Lobby | Office |
|-------|--------|-------|--------|
| 1     | 0.86   | 0.90  | 0.89   |
| 5     | 0.63   | 0.56  | 0.31   |

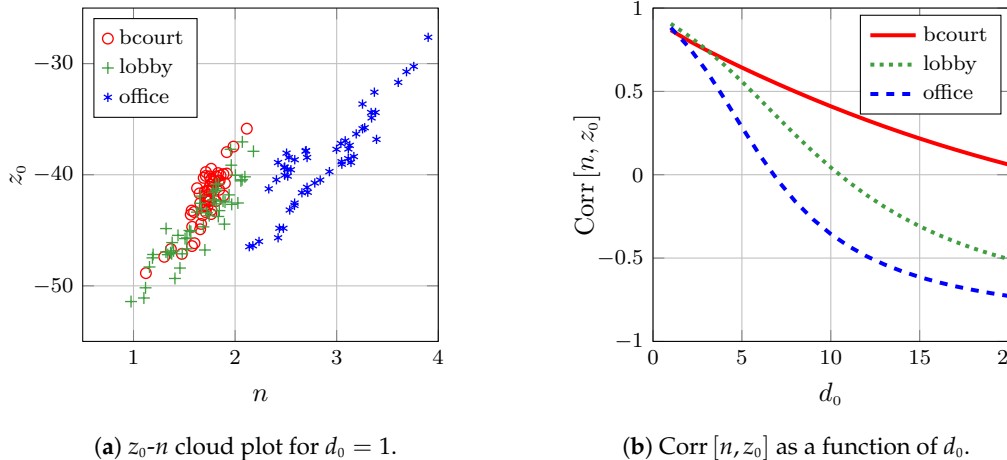

(**a**) $z_0$-$n$ cloud plot for $d_0 = 1$.  (**b**) Corr $[n, z_0]$ as a function of $d_0$.

**Figure 8.** Correlation between $z_0$ and $n$.

The change of sign in the correlation coefficient in the lobby and office is a consequence of the (concave) curvature of the RSS-distance scatter plots observed in those particular environments (see Figure 3) and the impossibility of the log-distance model to explain this trend. As argued in Section 4.2, the limitation of the log-distance model related to its linearity results in systematically biased RSS predictions. This bias is positive for small distance values and negative for large ones (see Figure 3). Additionally, the magnitude of the bias depends on the slope of the linear model fit to the data; that is, it depends upon $n$. Increments in $n$ will increase the bias for smaller $d$ values, and consequently also the intercept of the linear model along the y axis ($z_0$ for $d_0 = 1$ m). The same increments in $n$, on the other hand, will produce a negative bias for large enough distances. This explains the change of sign in the curves of the correlation in Figure 8b.

Figure 8b shows that the correlation depends on the type of environment. Interestingly, we can also see how the correlation tends to be similar in all of the environments for smaller $d_0$ values. Choosing $d_0 = 1$ resulted in a correlation of around 0.9 in all environments.

## 9. Discussion

The log-normal model is traditionally and pervasively used in the scientific literature related to WSNs assuming that its parameters are deterministic, and that $n$ and $z_0$ independently account for environmental and HW related factors, respectively. However, the empirical evidence presented in this article challenges these assumptions. First, the parameter's variability exposed in Section 5 suggests that these are better embodied by RVs rather than by deterministic ones. Furthermore, the strong correlation between $n$ and $z_0$ reported in Section 8 suggests that these two shall also be jointly distributed. This correlation reflects the interdependence between $n$ and $z_0$ induced by the model misspecification, and ultimately interrelates the effect of the HW and the environment in the parameters. Consequently, it is reasonable to claim that, with respect to the classic usage of the log-normal model, the RSS in WSN deployments is better represented by assuming the model parameters being RVs, where $n$ and $z_0$ shall be correlated jointly accounting for HW and environmental factors. The key concept addressed then is that of the representativeness of the model.

What are the implications of all this? How relevant is the parameters' variability? What are the potential benefits of taking it into account? How to do so in practice? Is it worth it? What are the consequences and risks of ignoring it? These all are relevant questions pertaining the significance of the results and claim presented, and need to be examined differently. While one can in principle qualitatively study the general implications based on arguments in relation to abstract concepts such as the model representativeness, the concrete assessment of the ultimate consequences and their relevance needs to be studied in a per-application basis taking into account the inference methods used and the goals pursued.

In particular, any sort of quantitative assessment requires first to make explicit decisions about what distributions the parameters follow, and then to explicitly incorporate these into the model and the algorithms/calculations that rely on it. Such concrete decisions and work are out of scope in this article, and consequently so are quantitative analyses of implications. Nevertheless, some general entailments can be brought up.

In essence, considering a measurement generative process that properly acknowledges, embraces and accounts for the parameter's variability opens the door to having a more representative model for the RSS in the whole WSN. This comes hand in hand with having (a) a smaller systematic bias in the overall predictions, (b) more informative measurements, and (c) more realistic predictions. This in turn opens the door to having more realistic conclusions in relation to, e.g., theoretical results (bounds on communication rates, variance of position estimators, etc.) and simulations that use properly generated synthetic data. All in all, properly accounting for this variability can reduce, to a limited extent, the model misspecification.

In this article, accounting for the parameters' variability has been explicitly done by using IMs to explain the data observed by each node individually, and ignoring it has been epitomized by the usage of one single model identified using the data collected by all nodes in each experiment/robot trajectory (the reference model). The gain in information for each measurement when switching from using the reference model to using IMs can be intuitively appreciated by analyzing the lowest panel of Figure 4. As one can see, and as expected, the values of $\sigma$ for reference models (thick dots) always clearly fall well over the mean of those for IMs, and consequently in the right tails of their associated empirical distributions. Thus, under the same customary independent and identically distributed (IID) observations assumption in Equation (3), using the reference models will result in each measurement from any node to carry exactly the same amount of information (essentially proportional to $1/\sigma^2$). On the other hand, adopting IMs makes measurements from different nodes to provide different amount of information according to the value of $\sigma$ from its respective IM. Moreover, because this value is on average smaller for IMs than for the reference model, the amount of information provided by the measurements from all nodes of the WSN will consequently be larger. Equivalently, one can verify that the log-likelihood of all the measurements (from all nodes) is larger when using IMs, and this is indeed the case for the data used in this article.

Another aspect that deserves attention in relation to the model representativeness is the impact of the parameters' variability when using CMs. As pointed out before, in that case the same pair of nodes is typically used to collect the training data in a certain AoI for the model identification. The resulting model is then used to predict the RSS of packets exchanged by all node pairs in the network independently of their position. However, the parameter's variability suggests that the resulting CM might not optimally explain the observations from nodes other than the pair used to collect training data and in the same AoI. Predictions associated to those other nodes might then incur in an extra bias due to this sub-optimal model representativeness. If a CM is then to be used, how to optimally estimate its parameters?

Intuitively, one can think of using training data collected with many different nodes and in many positions, so that the resulting model can better explain observations from all of them. The reference models introduced in Section 5 are the ultimate expression of this strategy, and are identified using data from all nodes. However, does using CMs identified with data from many nodes necessarily result in better predictions? In the area of localization, some work has been done in [9] to study this effect. Interestingly, it has been experimentally shown that a) the model parameters that minimize the localization error of single node position estimates were invariably different than the ones calculated using reference models, and b) using CMs identified with data from an increasing number of nodes can result in errors that gradually converge to those achieved when using a reference model, but that can vary substantially in the way attaining both, larger and smaller errors. Choosing the right model parameters that minimize the error seems to be then a question of luck, to some extent, and using models identified with data from many nodes results in more robust/less risky models, but note necessarily optimal.

Another approach to take into account the parameters' variability in the frame of localization is to jointly estimate the position and model parameters. This has been studied, e.g., in [15] also taking into account the spatial correlation of the shadowing due to the model misspecification. In this approach, the variability observed here was used to create priors for the model parameters, priors that guided a solver in selecting feasible parameter values during the ML optimization. Being the parameters part of the search space, this approach naturally allows them to vary depending on the experimental conditions.

The variability of the model parameters can also be taken into account in different ways in order to create more realistic synthetic data used in simulation-based studies. For example, in a setting comprising a series of fixed and mobile nodes, IMs can be used to generate the measurements associated to each communication channel involving a fixed and a mobile node. The parameters of the IMs can be drawn from prior distributions, perhaps calculated from real observations. In simulations of moderately dynamic settings, these parameters could be kept constant during the whole simulated experiment. Alternatively, for highly dynamic ones, one could also construct a time varying stochastic model of the parameters, and let them change over the virtual experiment.

As we can see, taking into account the natural parameter's variability characteristic of WSN deployments can be done in different ways. The underlying idea shall always be to better represent the measurements' generative process, and, when properly done, it is reasonable to expect that the outcomes of the inference done using the resulting models will also lead to more realistic conclusions with respect to assuming deterministic model parameters. While characteristic of our particular environment, the statistical values of the parameters' empirical distributions reported in Section 5 and the correlation between $n$ and $z_0$ presented in Section 8 can be used as an example of what can be found in a real, yet laboratory controlled, WSN deployment, and as a guideline to other studies.

## 10. Conclusions

The log-normal model is a mathematical empirical over-simplified and very convenient model of the RSS vs. the T-R distance. The price of this simplicity is its misspecification and consequent structural limitation when it comes to explaining real observations from some experimental settings, especially those exhibiting inhomogeneities. Its radiality renders it incapable of modeling the effect of, e.g., obstacles, an issue that is known in the literature to cause spatial correlation in the shadowing. Moreover, and as clearly shown in this article, the path-loss exponent and the reference power can be strongly correlated, an effect largely caused by the model linearity.

When identifying an empirical log-normal model using training data collected with different radio equipment, the model parameters can vary remarkably depending on the particular HW used and the peculiarities of the local surroundings. This becomes especially relevant in WSN deployments, typically characterized by HW variability and environmental inhomogeneity. The parameters are differently affected by these factors: HW variability mostly affects the reference power, whereas the environment has a stronger impact in the path-loss exponent. However, and as revealed by their correlation, they interact in a complex manner that makes it difficult to accurately account for their independent effects.

While $n$ is indeed mostly affected by the environment (being responsible for 68% of the relative change in IMs in one of our experiments), we have demonstrated that in practice it can be significantly affected by the antenna directivity (e.g., responsible for as much as 20% of relative change in one of our tests), and that the extent of this impact depends on the type of environment. While the average $n$ does not change much, the more open and homogeneous the environment is the more its dispersion will increase with the antenna directionality. Our experiments also demonstrated that increasing the antenna directionality can strongly increase the average value and dispersion of $\sigma$. The reference power, $z_0$, is directly affected by the directionality, and generally its dispersion increases with it as well.

The variability of the model parameters and their correlation suggest that real WSN deployments are better represented by a model with parameters embodied by random variables, rather than by independent deterministic ones. The strong correlation between $z_0$ and $n$ further implies that the classical assumption of independence between the impact of HW and environmental factors (the first ones affecting only $z_0$ and the latter only $n$) is not realistic in WSNs. Due to the intricacy of the interrelation between the model parameters, their (joint) distribution represents the combined effect of the particular HW being used and the peculiarities of the site of deployment.

Taking into account the parameters' variability in the measurement's generative process can partially reduce the model misspecification, leading in turn to having more informative measurement predictions, relatively less biased and more realistic in an aggregate scale. Consequently, it is reasonable to expect that the results derived from methods that acknowledge this variability will be also more realistic and plausible.

**Funding:** This work was partly supported by the Academy of Finland through the Center of Excellence in Generic Intelligent Machines Research (2008–2013), the main funding body that financed my dissertation, where this work is derived from.

**Acknowledgments:** The author would like to thank: Maurizio Bocca and Ossi Kaltiokallio for the preparation and handling of the WSN for the experiments, especially Ossi Kaltiokallio for their SW and the data collection. Jari Saarinen, main responsible of the design and construction of the robot, for his assistance with all the issues in relation to the robot. Matthieu Myrsky, for the implementation of the interfaces between the WSN and the robot and the handling of the robot during the experiments, and to all for their instructive comments.

**Conflicts of Interest:** The author declares no conflict of interest.

## Abbreviations

The following abbreviations are used in this manuscript:

| | |
|---|---|
| AoI | area of interest |
| CM | common model |
| HW | hardware |
| IID | independent and identically distributed |
| IM | individual model |
| LoS | line of sight |
| ML | maximum likelihood |
| OLS | ordinary least squares |
| PDF | probability distribution function |
| RSS | received signal strength |
| RV | random variable |
| T-R | transmitter–receiver |
| WSN | wireless sensor network |

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
