# Peer review of "Characterization of the Log-Normal Model for Received Signal Strength Measurements in Real Wireless Sensor Networks"

_jsan, doi:10.3390/jsan9010012_

Round 1

Reviewer 1 Report

This paper Is studding the structural limitation (known) of the classical RSS distance-log-normal-model. It provides evidence and tries to explain the reason (doing real experiments in 3 different scenarios) of the various model parameters and their correlation. The author argue that due to the varialiability of the model parameters – in different real WSN deployment – the usual log-normal model assuming independent deterministic parameter values can be outperformed by a joint probability distribution of the parameters.

This work is interesting and show good understanding of the path loos essence and the most known model used today. It fails though to clearly present the added value and innovation of the proposed model to compensate measured performance. It it important to extent related section on the results. It must be clear to the read what is the contribution of this work and the final result(s). Thus all related sections 8 and 9 must be extended significantly.

So more comments

The figures some times are difficult to read – Page 2/20 lines 68-71 must be revised, not clear. Page 3/20 lines 90-94, sentence must be revised – if this is the main outcome must be also more emphasized Section 2 experimental settings: What is the rate / length of the packets / transmit power – an eample figure could also be included. How the authors make sure that Tx power is enough to reach end of AoI. Section 2 – Do the authors assume simultaneous links? A usual case for WSNs – cite this situation in order to highlight the problem – i.e. https://www.mdpi.com/1099-4300/16/4/2085 Figure 2 – explain the reason for selecting these nodes-or provide more general results Section 4.1 – why WSN do not have omnidirectional antennas – provide reference or more explanation Section 4.1.1 : explain T-R angle (or provide reference) Figure 4 – must be explained and better described -, axis what is shows and how it provide to the content and added value of the paper

Author Response

Dear reviewer:

First of all, thanks for your constructive and instructive feedback. I have addressed all of your comments and taken the actions that I considered most appropriate to tackle them. As a result, I have made substantial corrections and additions as I deemed necessary based on your feedback.

I now proceed to answer your comments one by one.

* This work is interesting and show good understanding of the path loos essence and the most known model used today. It fails though to clearly present the added value and innovation of the proposed model to compensate measured performance. It it important to extent related section on the results. It must be clear to the read what is the contribution of this work and the final result(s). Thus all related sections 8 and 9 must be extended significantly.

A preliminary clarification: please note that I do not propose any particular model, as you seem to imply: that is not within the goals neither the intention in this article. A statistical model requires the full specification of all the density functions involved, and I do not do so. I simply show empirical evidence of the model parameters' variability, and provide the mean and standard deviation of the parameters' empirical distributions calculated from the experimental data. The aim is to show the variability of the parameters in a way that supports in a clear manner (visually) one of the main claims of the article: that WSNs are better represented by joint PDFs of the model's parameters rather than by independent deterministic values. But I do not make any statement about any PDF in particular. That is an issue to be decided on a per-deployment (and perhaps per-application) basis. Therefore, and as I explained before, I do not propose any model as such, neither I claim superiority of any model in particular. I simply show the limitations and inaccuracies in relation to how the log-normal model is typically used in the scientific community, and demonstrate that some classic assumptions are not realistic in WSNs.

In relation to the rest of the general comments provided by the reviewer: I have changed the text in the abstract, introduction and conclussions to better reflect the claims. I have also added a new section (Section 9 - Discussion) with a brief discussion about the potential implications and significance of the results. Please also note that it is not the intention to study or elaborate in detail about all the possible implications: that is a topic to be treated in further studies and articles.

I now continue with the answers to the particular comments.

* The figures some times are difficult to read

I assume that the reviewer refers mainly to figures 4 and 6. I have added more clear explanations about how to interpret Figure 4 and the information that it presents. Being Figure 6 of the same type and presenting similar information, the added information shall also ease its interpretation.

* Page 2/20 lines 68-71 must be revised, not clear.

The mentioned lines have been substantially modified, and the ideas conveyed clarified.

* Page 3/20 lines 90-94, sentence must be revised – if this is the main outcome must be also more emphasized

I have substantially modified the sentence, with more detailed explanations and clarifying/emphasizing its claim.

* Section 2 experimental settings: What is the rate / length of the packets / transmit power – an eample figure could also be included.

In relation to Section 2 (Experimental Setting), please note that, for the sake of brevity, I clearly state that the explanation about the setting is short. In fact, I only mention the points that I consider most important for the publication, since a complete explanation would unnecessarily long space. Also please note that I point the reader to another reference for more details. In order to make this more noticeable, I have moved that sentence to the beginning of the section.

Nevertheless, and answering to the particular questions from the reviewer:

The transmission rate is approximately 10 packets per second and node. This is already explained in Section 2.

Regarding the packet length: the payload of the packets sent by each node contains only the RSS-ID pairs corresponding to the packets sent by the neighbouring nodes in the previous communications round. Therefore, the length of the payload is variable between around 30 to around 50 bytes. In addition, the information sent over the air contains the headers and overhead of the underlying communication protocol.

The exact transmitted power is not known. We simply put all the nodes to transmit with full power. I have added this info (full power transmission) into Section 2.

Regarding the suggestion to add a figure; I do not fully understand what the reviewer would appreciate to see in the example figure requested. Is it about the robot, the environment, the position of the nodes and the robot trajectories? Is it about the details of the communication protocol and the content of the packets?

* How the authors make sure that Tx power is enough to reach end of AoI.

I don't: the power is kept constant at all times.

How to make sure that the Tx power is enough to completely cover an AoI is indeed a relevant question when one designs a wireless network aiming at ensuring continuous connectivity (e.g. between the robot and all the nodes at all times in my case). However, this is not the aim in the experiments done in relation to this article, neither I have claimed so. The idea is to move the robot in a certain area, area which I assume to have certain interest due to reasons pertaining the specific application, but not relevant to this article. For simplicity of the exposition, I chose to denote this area as the "area of interest", being this an abstraction that allows me to talk about the area in which the WSN is deployed without need to get into why.

For the sake of clarity in relation to this comment, I have added relevant information to Section 2. In particular, I have explicitly mentioned the use of a fixed Tx power, and I have clarified the meaning of the acronym AoI (see the corresponding footnote).

* Section 2 – Do the authors assume simultaneous links? A usual case for WSNs – cite this situation in order to highlight the problem – i.e. https://www.mdpi.com/1099-4300/16/4/2085

I assume that the reviewer denotes as "simultaneous links" what the referred article denotes as "uncoordinated links".

The communication protocol used followed a token-passing scheme strictly coordinated by a master node (essentially a form of TDMA), and thus there were no collision among packets sent by the different nodes. Furthermore, the channels used for communication were selected in each environment right before the experiments took place: only those that were free (as seen in a manual scan) were selected.

This all has been clarified in the text.

* Figure 2 – explain the reason for selecting these nodes-or provide more general results

The nodes corresponding to which the scatter-plots and histograms are displayed are chosen because the behavior of their data is typical/representative of that corresponding to nodes in the different environments. The emphasis is thus placed in the behavior of the data, not on specific nodes. It is not that I selected some nodes and then decided to display their data: I decided to display data showing the typical behavior in each scenario, and then I showed that data.

I believe that the text is clear in this sense. For example, in the sentence -"Figure 2 shows examples of typical \gls{RSS}-distance scatter plots and residual histograms together with their associated model for measurements collected by one node in each of our three environments"-. Also, the next sentence: -"Panel 2a shows a typical example for nodes in the basketball court"-. Moreover, the entire paragraph is written with emphasis on the general behavior, and referring to examples of models and/or data.

In my opinion, this essence is well captured in the text. Moreover, I consider that adding more explicit explanations about the reasons for selecting the nodes where the data comes from would be redundant and perhaps distractive, because, as explained, the emphasis is not on the nodes in particular, but in the behavior of the data.

* Section 4.1 – why WSN do not have omnidirectional antennas – provide reference or more explanation

I have now relaxed the sentence.

* Section 4.1.1 : explain T-R angle (or provide reference)

The text has been corrected. I have substituted the sentence "the relative T-R angle", which I agree uses an unclear terminology, by "the relative orientation between the transmitter and receiver".

* Figure 4 – must be explained and better described -, axis what is shows and how it provide to the content and added value of the paper

I have now substantially added more explanations in relation to Figure 4. In particular, I have added a more detailed explanation about how to read/interpret it, and the relevance of the information that it presents.

In essence, Figure 4 is a visual representation of the variability of the model parameters that one can experience in a real WSN deployment. The results that it presents (including that of the variability) is something that I have not seen in any other study in the scientific literature yet, in terms of the richness of the data used to produce it (using a mobile robot) and the visualily of the results.

Reviewer 2 Report

This paper addresses the characterization of the log-normal model of RSSI in WSN. The experimental setup and discussion of results look valid. However, it is not clearly presented what original contributions are, in particular, in abstract, introduction and conclusion. 

Therefore, it is recommended to improve the presentation of implications and originality of the experimental results.

Author Response

Dear reviewer:

First of all, thanks for your constructive and instructive feedback. I now proceed to answer your comments.

* This paper addresses the characterization of the log-normal model of RSSI in WSN. The experimental setup and discussion of results look valid. However, it is not clearly presented what original contributions are, in particular, in abstract, introduction and conclusion.
Therefore, it is recommended to improve the presentation of implications and originality of the experimental results.

The Abstract and Introduction have been substantially modified in order to better reflect the aim of the article and clarifying the knowledge gap it tries to fill. In addition, a new section has been added (Section 9 - Discussion) specifically tackling the implications and significance, and also clarifying the scope of the article with respect to these.